# Contactin-1 links autoimmune neuropathy and membranous glomerulonephritis

**Janev Fehmi**[1☉]*, **Alexander J. Davies**[1☉], **Marilina Antonelou**[2], **Stephen Keddie**[3], **Sonja Pikkupeura**[1], **Luis Querol**[4], **Emilien Delmont**[5], **Andrea Cortese**[3,6], **Diego Franciotta**[7], **Staffan Persson**[8], **Jonathan Barratt**[9], **Ruth Pepper**[2], **Filipa Farinha**[10], **Anisur Rahman**[10], **Diana Canetti**[11], **Janet A. Gilbertson**[11], **Nigel B. Rendell**[11], **Aleksandar Radunovic**[12], **Thomas Minton**[13], **Geraint Fuller**[14], **Sinead M. Murphy**[15], **Aisling S. Carr**[3], **Mary R. Reilly**[3], **Filip Eftimov**[16], **Luuk Wieske**[16], **Charlotte E. Teunissen**[16], **Ian S. D. Roberts**[17], **Neil Ashman**[18], **Alan D. Salama**[2‡], **Simon Rinaldi**[1‡]

1 Nuffield Department of Clinical Neurosciences, University of Oxford, John Radcliffe Hospital, Oxford, United Kingdom, 2 University College London Department of Renal Medicine, Royal Free Hospital, London, United Kingdom, 3 Centre for Neuromuscular Disease, National Hospital of Neurology and Neurosurgery and Department of Neuromuscular Diseases, UCL Institute of Neurology, Queen Square, London, United Kingdom, 4 Neuromuscular Diseases Unit, Department of Neurology, Hospital de la Santa Creu i Sant Pau, Universitat Autònoma de Barcelona, Barcelona, Spain, 5 Referral Centre for ALS and Neuromuscular Diseases, Hospital La Timone, Marseille, France, 6 Department of Brain and Behaviour sciences, University of Pavia, Pavia, Italy, 7 IRCCS, Mondino Foundation, Pavia, Italy, 8 Faculty of Medicine, Department of Clinical Sciences Lund, Neurology, Lund University, Lund, Sweden, 9 Department of Cardiovascular Sciences, University of Leicester, Leicester, United Kingdom, 10 Centre for Rheumatology and Bloomsbury Rheumatology Unit, Division of Medicine, University College London, London, United Kingdom, 11 Wolfson Drug Discovery Unit and National Amyloidosis Centre, Centre for Amyloidosis and Acute Phase Proteins, Division of Medicine, University College London, London, United Kingdom, 12 Barts Neuromuscular Diseases Centre, Royal London Hospital, London, United Kingdom, 13 Institute of Clinical Neurosciences, University of Bristol, Bristol, United Kingdom, 14 Department of Neurology, Gloucestershire Royal Hospital, Gloucester, United Kingdom, 15 Department of Neurology, Tallaght University Hospital & Academic Unit of Neurology, Trinity College, Dublin, Ireland, 16 Department of Neurology and Neurophysiology, Amsterdam Neuroscience, Amsterdam UMC, Location AMC, Amsterdam, The Netherlands, 17 Department of Cellular Pathology, John Radcliffe Hospital, Oxford, United Kingdom, 18 Barts Renal Unit, The Royal London Hospital, London, United Kingdom

☉ These authors contributed equally to this work.
‡ ADS and SR also contributed equally to this work.
* Janev.fehmi@ndcn.ox.ac.uk

**Data Availability Statement:** All relevant data are within the paper and its Supporting Information files.

**Funding:** This work was supported by a GBS|CIDP Foundation International Benson Fellowship

## Abstract

Membranous glomerulonephritis (MGN) is a common cause of nephrotic syndrome in adults, mediated by glomerular antibody deposition to an increasing number of newly recognised antigens. Previous case reports have suggested an association between patients with anti-contactin-1 (CNTN1)-mediated neuropathies and MGN. In an observational study we investigated the pathobiology and extent of this potential cause of MGN by examining the association of antibodies against CNTN1 with the clinical features of a cohort of 468 patients with suspected immune-mediated neuropathies, 295 with idiopathic MGN, and 256 controls. Neuronal and glomerular binding of patient IgG, serum CNTN1 antibody and protein levels, as well as immune-complex deposition were determined. We identified 15 patients with immune-mediated neuropathy and concurrent nephrotic syndrome (biopsy proven MGN in 12/12), and 4 patients with isolated MGN from an idiopathic MGN cohort, all seropositive for IgG4 CNTN1 antibodies. CNTN1-containing immune complexes were found in the renal

awarded to JF (1709HM001/SB17) (https://www.gbs-cidp.org/) and a Medical Research Council UK Clinician Scientist Fellowship (MR/P008399/1) (https://www.ukri.org/councils/mrc/) awarded to SR. AJD is supported by a UKRI Future Leaders Fellowship (MR/V02552X/1) and a Human Immune Discovery Initiative award funded by the NIHR Oxford Biomedical Research Centre. CT received support from the EU/EFPIA Innovative Medicines Initiative Joint Undertaking (EMIF grant 115372) (https://european-union.europa.eu/institutions-law-budget/institutions-and-bodies/institutions-and-bodies-profiles/innovative-health-initiative-joint-undertaking-ihi-ju_en). AC thanks the Medical Research Council (MR/T001712/1), Fondazione CARIPLO (2019-1836), Italian Ministry of Health (Ricerca Corrente 2020), and the Inherited Neuropathy Consortium (INC) for grant support. ASC received funding from NIHR University College London Hospitals Biomedical research centre. MA is a clinical fellow with grant reference number MR/P001777/1. FF was supported by LUPUS UK. The funders had no role in study design, data collection and analysis, decision to publish, or preparation of the manuscript.

**Competing interests:** SR runs a not-for-profit diagnostic testing service for nodal/paranodal antibodies. All other authors have declared that no competing interests exist. This does not alter our adherence to PLOS ONE policies on sharing data and materials.

glomeruli of patients with CNTN1 antibodies, but not in control kidneys. CNTN1 peptides were identified in glomeruli by mass spectroscopy. CNTN1 seropositive patients were largely resistant to first-line neuropathy treatments but achieved a good outcome with escalation therapies. Neurological and renal function improved in parallel with suppressed antibody titres. The reason for isolated MGN without clinical neuropathy is unclear. We show that CNTN1, found in peripheral nerves and kidney glomeruli, is a common target for auto-antibody-mediated pathology and may account for between 1 and 2% of idiopathic MGN cases. Greater awareness of this cross-system syndrome should facilitate earlier diagnosis and more timely use of effective treatment.

## Introduction

Peripheral neuropathy and renal disease commonly co-occur. In some cases, neuropathy may be secondary to uraemia, micronutrient deficiencies or the imbalanced metabolic milieu of renal failure [1]. Other causes include diabetes, haematological disorders such as lymphoma or myeloma [2], and drugs or metabolites that are both nephro- and neuro-toxic [3, 4]. Genetic neuropathies, such as those associated with Fabry disease [5] and Charcot-Marie-Tooth dominant-intermediate type E [6], can also be complicated by proteinuria and progressive renal failure. Accurate identification of the underlying cause is crucial for guiding management.

Membranous glomerulonephritis (MGN) is one of the most common causes of nephrotic syndrome in adults and is strongly associated with autoantibodies to kidney antigens [7–9]. Previous small series and case reports have suggested an association between nephrotic syndrome and inflammatory neuropathies, namely Guillain-Barré syndrome (GBS) or chronic inflammatory demyelinating polyneuropathy (CIDP). However, the mechanisms linking these conditions have remained unclear. More recently, this combined presentation has been described in some patients with nodal or paranodal antibodies [10–13]. It has been speculated that this is due to a common autoimmune process involving both the peripheral nerve and kidney. Here, we demonstrate that antibodies targeting contactin-1 (CNTN1), a neuronal membrane protein anchoring paranodal myelin to the underlying axon, mechanistically connect these pathologies, in the largest cohort to date and identify a distinct and treatable neuro-renal syndrome. In addition, we show that a small percentage of idiopathic MGN may be caused by anti-CNTN1 antibodies, without overt neuropathy, and confirm that CNTN1 peptides are expressed in the affected glomeruli while RNA expression has been demonstrated in normal glomeruli, adding CNTN1 to the list of other important MGN antigens.

## Materials and methods

### Patient cohort and samples

From January 2015 to August 2019 we prospectively recruited patients attending the peripheral nerve clinic in the John Radcliffe Hospital (Oxford, UK) with either confirmed or suspected inflammatory neuropathy to an observational study (Research Ethics Committee approval number 14/SC/0280). These patients provided informed written consent. Serum samples from these patients, and patients with suspected inflammatory neuropathies, received by our laboratory for diagnostic testing between August 2017 and August 2019. were screened for antibodies against paranodal (CNTN1, contactin-associated protein 1—Caspr1, neurofascin 155—NF155) and nodal (NF140/186) antigens.

To investigate whether CNTN1 antibodies might be more widely associated with nephrotic syndrome caused by idiopathic MGN itself, we examined 295 serum samples from patients with idiopathic membranous nephropathy, collected as part of the MRC Glomerulonephritis bank [14].

Serum samples from 70 patients with other antibody-mediated CNS neurological disorders, 20 with multiple sclerosis, 120 individuals without neurological disease, and 46 patients with lupus nephritis, including pure class V membranous lupus nephritis, were also obtained as controls (S1 Fig).

Information that could identify individual participants was stored confidentially and only accessible at the time of data collection to treating physicians, or those with necessary ethical approval and Good Clinical Practice (GCP) training. Data was subsequently de-identified.

**Nodal/paranodal antibody testing and binding to peripheral nerve.** All sera were initially screened using a live, transiently transfected cell-based assay (CBA), as previously described with slight modification [12]. Positive results, as well as titre and IgG subclass, were confirmed using ELISA.

**Cell based assay (CBA).** Human embryonic kidney 293T (HEK) cells were either co-transfected with human DNA plasmid constructs for CNTN1 and Caspr1, NF155, or NF186 followed by incubation with patient sera. Fluorescently tagged secondary and tertiary antibodies against human IgG or human IgG subclasses 1–4 were used to visualise cell membrane binding by an investigator blind to the sample identity.

**Myelinating co-cultures.** Sera were assessed for topographical binding of IgG using myelinated co-cultures, generated using sensory neurons derived from human induced pluripotent stem cells (iPSC), according to a previously published protocol [15, 16]. For immunolabelling, live co-cultures were incubated with patient sera then fixed prior to labelling with secondary antibodies against human IgG.

**Teased nerve fibres.** Single fibres from 4% paraformaldehyde-fixed mouse sciatic nerve were teased on glass sides, air-dried and permeabilised with 100% acetone (10 min at –20˚C), and blocked with 5% fish gelatin and 0.1% triton-X in PBS. They were incubated overnight at 4˚C with guinea pig contactin-associated protein 1 (Caspr1) antibodies (1:1000) to label the paranodes, and patient sera (1:200). Immunolabelling was visualised with fluorescent-conjugated secondary antibodies.

**Human kidney CNTN1 immunohistochemistry.** Tissue sections of kidney biopsies from patients 1, 2, 12 and 13 (S1 Table in S1 File), as well as healthy human kidney, were deparaffinised, rehydrated and underwent antigen retrieval by boiling in citrate buffer. After permeabilisation and blocking (0.3% triton-X 100, 10% normal donkey serum in PBS), sections were incubated overnight at 4˚C with goat anti-CNTN1 (1:800; 2.5 μg/ml) (AF904, R&D systems), followed by a biotinylated anti-rabbit secondary antibody. Endogenous peroxidase was quenched with 3% hydrogen peroxide prior to labelling with a horseradish peroxidase-streptavidin complex (Vectorstain Elite ABC HRP kit, Vector, PK-6100) and developed with diaminobenzidine (DAB) reagent (SK-4100, Vector) followed by haematoxylin counterstain. Sections of healthy human cortex were used as positive control.

## Laser microdissection and proteomics

Formalin-fixed paraffin embedded renal or human brain tissues, were laser micro-dissected and captured using the Leica LDM7 system, and analysed in a proteomics approach [17] by LC-MS/MS using a Thermo Scientific Q-Exactive Plus mass spectrometer. MS raw data were analysed by Mascot software (Matrix Science, London, UK) using the Swiss-Prot human database.

### Pre-adsorption of sera with CNTN1 protein

Sera available locally (from patients 1–3 and 12, 13 and 15 S1 Table in S1 File) were pre-adsorbed at 1:100 dilution with 1μg of CNTN1 protein overnight at 4˚C and reapplied for testing in the CBA and against myelinated co-cultures to confirm CNTN1 as the specific antibody target.

### Serum CNTN1 protein measurement

Serum CNTN-1 protein levels were measured on the Luminex® platform according to the manufacturer's instructions (Human Magnetic Luminex Assay, R&D systems) as previously described [18]. Samples were coded randomly and analysed in duplicate. Measurements with a coefficient of variation >15% and outliers were repeated.

### Serum immune complex precipitation

Whole sera were treated with polyethylene glycol (PEG) 6000 to a final concentration of 2.5% overnight at 4˚C. Immune complexes were precipitated as previously described [19], by centrifugation at 2000g for 30 minutes at 4˚C, resuspending pellets in 30μl of prewarmed PBS, and analysed using SDS-PAGE and western blotting.

### Statistics

Statistical analysis was performed using Prism 8 (version 8.0.2). Statistical significance for contingency data was assessed by Fishers exact test. Differences between CNTN1 antibody titres in remission versus in active disease, and in serum CNTN1 levels in the presence or absence of nephrotic syndrome, were assessed by the Mann-Whitney test. A p-value of <0.05 was considered significant.

## Results

### Contactin-1 antibodies serologically link immune-mediated neuropathies to membranous glomerulonephritis causing nephrotic syndrome

We prospectively screened 468 patients with suspected inflammatory neuropathies for antibodies against nodal and paranodal targets (CNTN1, Caspr1, neurofascin 155 –NF155, neurofascin 186 –NF186) using a live, cell-based assay (see methods online). Thirty-six (7.7%) patients were seropositive for antibodies targeting one of the nodal/paranodal proteins. Ten patients (2.1% of the total cohort and 27.8% of the seropositive group) were monospecific for CNTN1 antibodies (S1 Fig), and eight of these additionally had nephrotic syndrome. Among the 432 seronegative patients, 147 (34%) underwent investigation of renal function; in contrast with the seropositive group, only 5/147 (3.4%) of these had features supporting a diagnosis of nephrotic syndrome; (p<0.001, Fisher's exact test, OR for nephrotic syndrome with CNTN1 antibodies versus seronegative patients = 113.6, 95% CI 11.2 to 169.1). Two patients with antibodies against other nodal/paranodal targets were diagnosed with nephrotic syndrome.

Subsequently, a further seven patients with CNTN1 antibodies, an immune-mediated neuropathy, and confirmed nephrotic syndrome or suspected nephrotic syndrome with MGN confirmed on biopsy, were retrospectively identified from international collaborating centres. IgG4 was the predominant subclass, except in four patients, where IgG1 was equally reactive (S2 Fig). Thus a total of 15 patients, summarised here (S1 Table in S1 File) had a CNTN1 antibody-associated neuropathy and nephropathy.

We also tested for CNTN1 antibodies in 295 patients with idiopathic MGN without neuropathy. Four (1.4%) were seropositive (S1 Fig). PLA$_2$R antibody testing was performed where possible, and found to be negative in these four, and 10/15 patients with CNTN1 antibodies, nephrotic syndrome and a neuropathy (14/19 in total)—the remaining sera unavailable for further testing. Testing for other newly described MGN antigens was not done due to lack of clinical assay availability.

CNTN1 antibodies were not detected in 256 control sera (90 from patients with other neurological diseases, 46 from patients with lupus nephritis, 12 of which had a pure class V membranous lupus nephritis, and 120 from individuals with other non-neurological, non-renal diseases).

## CNTN1-containing immune complexes are found in the glomeruli of patients with CNTN1 antibodies

Renal biopsies were performed in 12 of the 15 patients. In all cases biopsies were characteristic of membranous glomerulonephritis, with diffuse thickening of glomerular capillary walls, and basement membrane spikes and lucencies evident on silver stain. Where available, immunostaining revealed glomerular capillary wall IgG (Fig 1A) and complement C3 (not shown), with subepithelial electron dense deposits, representative of immune complexes, on electron microscopy (Fig 1B). Granular deposition of CNTN1 protein was confirmed along glomerular basement membranes by immunohistochemistry (Fig 1C and S5 Fig). Furthermore, laser dissection of glomeruli followed by LC-MS/MS proteomic analysis revealed a 10 amino acid peptide (p.617-626, ATSVALTWSR) in CNTN1-immunopositive renal tissue, which matched with recombinant CNTN1 protein (S6 Fig).

In contrast, no CNTN1 staining was observed in glomeruli from healthy donor kidney tissue (not shown) or PLA$_2$R-associated MGN patient kidney tissue (Fig 1D). However, CNTN1 mRNA was detectable in healthy human kidney by polymerase chain reaction (PCR), and low-level CNTN1-protein expression could be observed by western blot (S7 Fig).

## Low serum CNTN1 protein levels correlated with immune complex formation

We have recently shown that serum levels of CNTN1 protein (sCNTN1) are lower in antibody-positive neuropathy patients compared with seronegative and healthy controls [18]. Consistent with this, we found sCNTN1 levels were significantly lower at presentation in CNTN1-antibody positive patients (median 1.02 ng/ml, range 0.02 to 36.37) compared to 25 randomly selected seronegative patients with CIDP (four of whom had concurrent nephrotic syndrome) (median 10.67 ng/ml, range 3.96 to 30.06) and 25 seronegative patients with nephrotic syndrome only (median 13.91 ng/ml, range 4.37 to 26.72) (p = 0.003 and p<0.001, Kruskal-Wallis) (Fig 2A). Overall, there was a non-significant trend for sCNTN1 and CNTN1 antibody levels to be inversely correlated (p = 0.099, R2 = 0.1088, simple linear regression) (Fig 2B). Finally, using western blotting we observed CNTN1 within the PEG precipitates isolated from the serum of a patient with active renal and neurological disease (patient 14 in S1 Table in S1 File). CNTN1 was not detected within the PEG precipitates isolated from the serum of the same patient during remission, or in that of a healthy control donor (Fig 2C and S8A and S8B Fig) suggesting the association of CNTN1 with immune complexes at disease peak. Conversely, CNTN1 was undetectable in the soluble fraction of PEG-treated serum of P14 during active disease (S8A and S8B Fig), consistent with the low levels of serum CNTN1 in the other anti-CNTN1 antibody patient samples (Fig 2A).

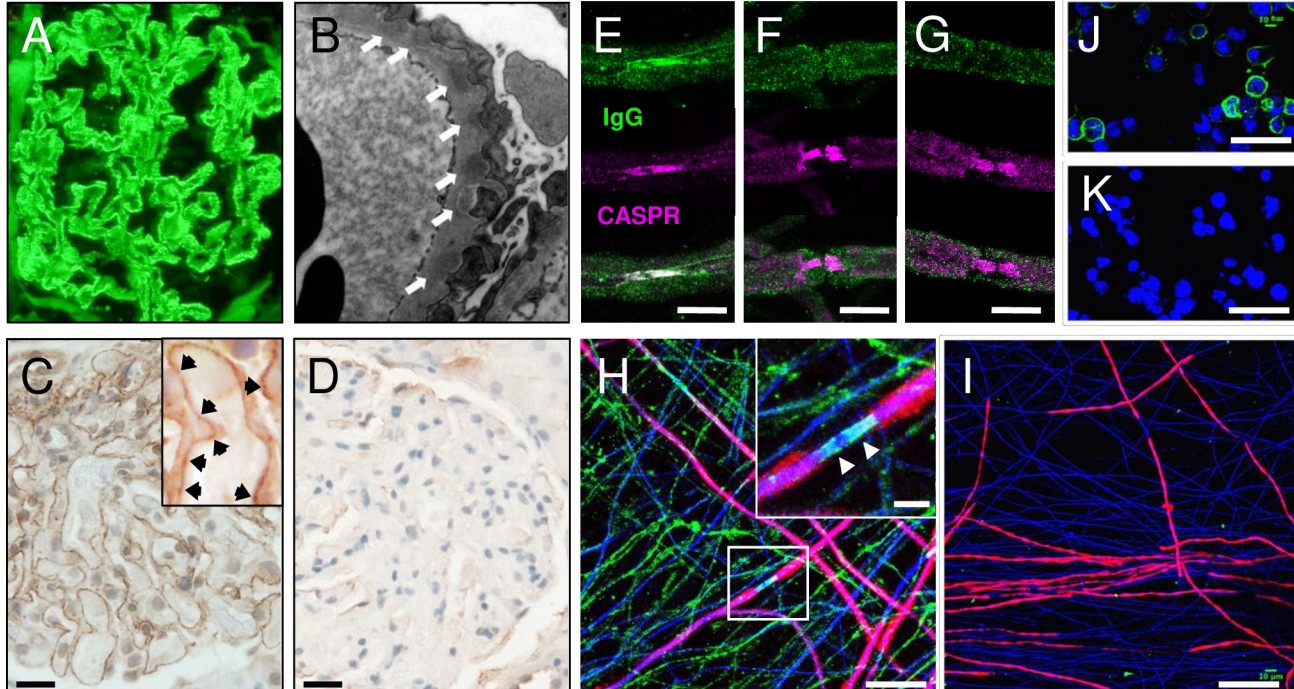

**Fig 1. Contactin-1 immunoreactivity in kidney immune deposits and peripheral nerve tissue.** A) Immunofluorescence of IgG deposition along glomerular capillary wall in anti-CNTN1 positive MGN patient kidney (P12 from S1 Table in S1 File). B) Electron microscopy reveals immune complex deposits on the extracapillary side of the glomerular basement membrane, visible as electron dense material (white arrows). C) Diaminobenzidine immunohistochemistry for CNTN1 is observed along the basement membrane of glomeruli from the same anti-CNTN1 positive MGN patient. Higher magnification image of a glomerulus demonstrating enrichment of CNTN1 in deposits lining the thickened basement membrane, characteristic of primary MGN (inset, black arrows). Equivalent results were obtained from biopsies of three further patients (S5 Fig). D) No CNTN1 labelling was observed in biopsy sections from an anti-PLA$_2$R-positive patient with MGN. Scale bars in C-D, 20 μm. E-G) Labelling of the paranode in teased fibres from mouse sciatic nerve with IgG (green) from pre-treatment anti-CNTN1-positive human sera (E) is abrogated post-treatment (F) and absent in healthy control sera (G). Bound human IgG co-localises with CASPR at the paranode (magenta). Scale bars, 10μm. H) Human IgG binding (green) to myelinated co-cultures treated with patient sera sampled before steroid treatment. Cultures were permeabilised with methanol prior to sera incubation. Note diffuse axonal pattern and paranodal localisation of human IgG (green) (arrowheads in inset). Myelin basic protein (MBP, red) indicates myelin, and neurofilament (NF200, blue) indicates axons. Scale bar, 25μm (5μm in inset) I) Loss of human IgG binding (green) to myelinated co-cultures treated with patient sera sampled after steroid treatment. Scale bar 50 μm. J, K) CNTN1-transfected HEK293 cells (CBA) treated with patient sera sampled before (J) and after (K) steroid treatment. Serum titration in CNTN1 CBA shows a fall of anti-CNTN1 titre from 1:6400 to negative subsequent to steroid treatment. Scale bars, 50 μm.

## CNTN1 antibodies from seropositive patients target peripheral nerves and paranodes

Patient serum showed IgG reactivity with teased mouse nerve fibres, colocalising with Caspr1 at the paranodes (Fig 1E–1G). IgG bound to unmyelinated axons in live human stem cell-derived sensory neuron co-cultures (Fig 1H), as well as paranodal regions of myelinated axons (Fig 1H, *inset*). Following immunosuppressive treatment which led to complete suppression of CNTN1 antibody titres on the transient transfection cell-based assay (Fig 1J and 1K), all IgG labelling was lost from neuronal axons (Fig 1I).

Further, locally available sera from patients found to have CNTN1 antibodies, neuropathy and nephrotic syndrome were pre-adsorbed with recombinant CNTN1 protein and re-applied to live CBA or neuronal co-cultures (S3 and S4 Figs). Immunolabelling was markedly reduced for all pre-adsorbed sera tested in both assays, confirming CNTN1 as the antigen for serum IgG binding in the live cell culture systems.

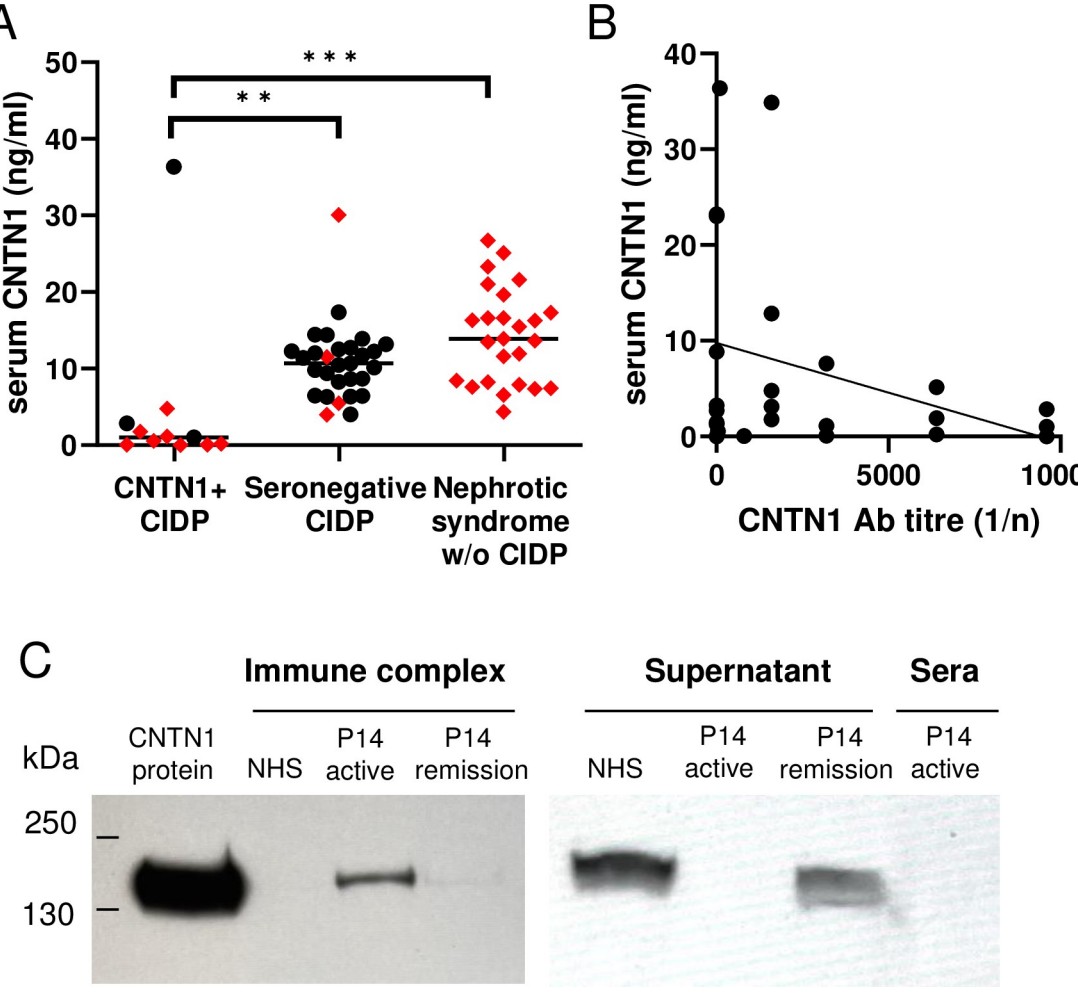

**Fig 2. Fall is serum CNTN1 protein is associated with immune complex formation in active disease. A)** Serum CNTN1 (sCNTN1) protein levels are significantly lower in CNTN1 antibody seropositive CIDP (n = 11; sera available for testing from 10 patients in prospective group and 1 in retrospective group) vs seronegative CIDP patients (n = 29), and those with nephrotic syndrome but without CIDP (n = 25). Red diamonds and black dots indicate patients with and without nephrotic syndrome, respectively. Although 9/11 patients from the CNTN1 Ab+ CIDP group developed MGN, at the time of testing for sCNTN1 only 8/ 11 had nephrotic syndrome. Kruskal-Wallis p<0.001 overall, with post-test pairwise comparisons (CNTN1 Ab+ vs. seronegative CIDP, p = 0.003, vs. nephrotic syndrome alone p<0.001, seronegative CIDP vs. nephrotic syndrome ns, p = 0.19). **B)** Overall, there was a non-significant trend for sCNTN1 and CNTN1 antibody levels to be negatively correlated (p = 0.099, R2 = 0.1088, simple linear regression). However in 5/8 patients where CNTN1 antibodies later became undetectable, sCNTN1 remained low. **C)** Western blot for CNTN1 in the insoluble fraction of polyethylene glycol (PEG) treated healthy control normal human serum (NHS) and serum of patient 14 at nadir of active disease compared to remission. Recombinant CNTN1 protein (0.003 μg) loading; 5μl immune complexes purified from 300μl of patient serum. A repeat blot reveals reciprocal CNTN1 bands detectable only from the soluble fraction of the NHS and P14 serum in remission, but not during nadir of active disease (5μl loaded).

## Patients with CNTN1 antibodies have a distinct clinical syndrome

The majority of patients with CNTN1 antibodies presented with a neuro-renal syndrome characterised by a rapidly progressive, disabling, sensory-motor neuropathy accompanied by typical nephrotic syndrome, with oedema, low serum albumin, and elevated levels of urinary proteinuria. Additional features, including pain, tremor and autonomic dysfunction, were common, and over one-quarter had a prior diagnosis of diabetes. A summary of clinical characteristics is provided in Table 1, and further detail in S1 Table in S1 File. Smaller numbers of patients had isolated kidney or nerve involvement.

**Table 1. Summary of patients with neuropathy, nephrotic syndrome and CNTN1 antibodies.**

| | SUMMARY OF KEY CHARACTERISTICS | | |
|---|---|---|---|
| | Characteristic | | Patients % (No.) |
| **CLINICAL** | Age (median, range) | | 59 (39–79) |
| | Male | | 80 (12/15) |
| | Diabetes | | 26.7 (4/15) |
| | Acute/Subacute onset | | 86.7 (13/15) |
| | Additional clinical features (inc ataxia, tremor, cranial nerve palsy, autonomic dysfunction and abdominal weakness) | | 93.3 (14/15) |
| | Pain | | 53.3 (8/15) |
| | Median nadir disability—mRS (range) | | 5 (2–6) |
| **INVESTIGATIONS** | CSF | Median protein g/l (range) | 1.99 (0.24–5.93) |
| | | Normal protein | 21.4 (3/14) |
| | | WCC >5 | 28.6 (4/14) |
| | Meets electrodiagnostic criteria for CIDP [A] | | 93.3 (14/15) |
| | Median serum albumin g/l (range) | | 22 (11–36) |
| | Median urinary PCR mg/mmol (range) | | 1032 (633–4638) |
| | Median urinary protein g/24hr (range) | | 3.35 (2.2–10) |
| | Clinical criteria met for nephrotic syndrome [B] | | 92.3 (12/13) [C] |
| | MGN on renal biopsy | | 100 (12/12) |
| | Complement (C3 or C1q) deposition noted | | 58 (7/12) |
| | PLA$_2$R negative (IHC or serum antibodies) | | 100 (10/10) |
| **TREATMENT RESPONSE [D]** | First line therapy (Steroids +/-IVIg +/- PE) | Responders | 20 (3/15) |
| | | Non-responders | 66.7 (10/15) |
| | Escalation therapy [E] (inc. RTX) | Responders | 73 (8/11) |
| | | Non-responders | 27 (3/11) |
| **OUTCOME** | mRS (median, range) | | 1 (0–6) |
| | Complete remission (of both neuropathy/nephropathy) | | 31 (5/15) |
| | Significant improvement (≥ 2 points mRS) | | 38 (6/15) |
| | Minimal improvement/stabilisation (of both neuropathy/nephropathy) | | 6 (1/15) |
| | Death | | 25 (3/15) |

[A] As per EFNS/PNS [20]

[B] One or more of urinary PCR >300mg/mmol, proteinuria >3g/24 hours, serum albumin <35g/l

[C] 1 patient did not meet clinical criteria for nephrotic syndrome, and 2 did not have nephrotic investigations available. All 3 of these had biopsy proven MGN.

[D] Definition of treatment response

*Responder*: Complete remission, Sustained functional improvement > = 2 points mRS

*Non-responder*: Death, no response, transient response or improvement by only 1 point on the mRS

[E] Escalation therapy includes (in order of frequency used): Rituximab, Cyclophosphamide, Azathioprine, Tacrolimus, Autologous stem cell transplant, Mycophenolate Mofetil, Immunoadsorption

Abbreviations: CIDP–chronic inflammatory demyelinating polyneuropathy, CSF–Cerebrospinal fluid, IHC–Immunohistochemistry, IVIg–Intravenous immunoglobulin, mRS–Modified Rankin score, PE–Plasma exchange, PCR–Protein:Creatinine ratio, RTX–Rituximab, WCC–White cell count.

## The nephropathy and neuropathy are immunotherapy responsive and improve in parallel with a reduction in CNTN1 antibody titres

Overall, 75% of patients achieved either complete remission, significant improvement, or stabilisation of their neuropathy and nephropathy in parallel. However, only 20% (3/15) responded well to the first-line treatments typically used for inflammatory neuropathies (steroids, IVIg, plasmapheresis) (Table 1). Four different treatment modalities, ultimately

including escalation immunotherapies such as rituximab (RTX), were typically required to achieve a response. Two patients (noticeably the eldest of the cohort) died from multi-organ or respiratory failure whilst on the intensive care unit. A third patient died due to complications from coronavirus disease-19 (COVID-19) following prior treatment with IVIg, steroids and rituximab.

CNTN1 antibody titres were high prior to immunosuppressive treatment, and lower in remission of both neuropathy and nephropathy (Fig 3A) (p = 0.001). In all 6 patients for whom both pre- and post-treatment serum samples were available, anti-CNTN1 titres fell in parallel with a reduction on the CIDP Disease Activity Scale (CDAS, Fig 3B). Serial assessments in one patient showed that improvements in strength and disability paralleled the fall in CNTN1 antibody titres, and a fall in urinary protein levels followed shortly after (Fig 3C).

## Discussion

This study identifies CNTN1 antibodies as the pathological link between immune-mediated neuropathies and nephrotic syndrome caused by MGN, providing a mechanistic explanation for numerous previous clinical observations [10–13, 21]. We also demonstrate a significant association between CNTN1 antibody titres and immunotherapy responsiveness in this neuro-renal syndrome, and establish CNTN1 as another new antigenic target in a small proportion of "idiopathic" membranous glomerulonephritis. A previous report found non-concordant presentation of nephrotic syndrome and neuropathy in a patient with anti-CNTN1 antibodies—separated by a year, implying that some patients may further develop the combined neuro-renal syndrome at a later stage [22].

Without treatment, up to half of patients with MGN develop end-stage renal failure [23]. In keeping with recent data, patients with CNTN1 antibodies were poorly responsive to the typical first line therapies used for inflammatory neuropathies [24]. However, escalation therapies substantially reduced disability overall, and the associated renal disease invariably improved in parallel, as has recently been described in one further case report [25].

Around 70% of patients with idiopathic MGN have detectable serum IgG4 autoantibodies against the glomerular podocyte antigen phospholipase $A_2$ receptor ($PLA_2R$) [7]. As such, testing for $PLA_2R$ antibodies has transformed the management of MGN, in many cases potentially avoiding the need for diagnostic biopsy. Similarly, the early detection of CNTN1 antibodies may negate the need for further renal investigation in anti-$PLA_2R$ negative patients, and provide an opportunity for earlier, therapeutic intervention. In addition, falling anti-CNTN1 titres mirrored clinical improvement, suggesting their potential as a prognostic marker. Indeed, high titre $PLA_2R$ antibodies predict a worse prognosis, [26] with reduced titres found to precede the onset of remission [27]. As with anti-$PLA_2R$ MGN [28], we found the detection of serum CNTN1 antibodies often preceded the onset of proteinuria- confirming their utility as biomarkers.

Renal biopsy still needs to be considered for seronegative patients in whom antibody-mediated nephrotic syndrome is suspected, as evidenced by the fact that we observed one patient to have glomerular CNTN1-immune-complex deposition (patient 2 in S5 Fig) despite having become CNTN1 seronegative (following steroid treatment) by the time of renal biopsy.

We confirm the presence of CNTN1 in both renal and peripheral nerve tissue, and provide evidence that CNTN1 antibodies identify an aberrant immune response targeting both peripheral nerve and kidney. Our observation of low-level expression of CNTN1 protein in healthy kidney is supported by recent single cell transcriptomic profiling of human kidney, which identified CNTN1 in podocytes [29], while others have recently described CNTN1 expression in normal kidney glomeruli and co-localisation of CNTN1 and IgG4 in the glomerular

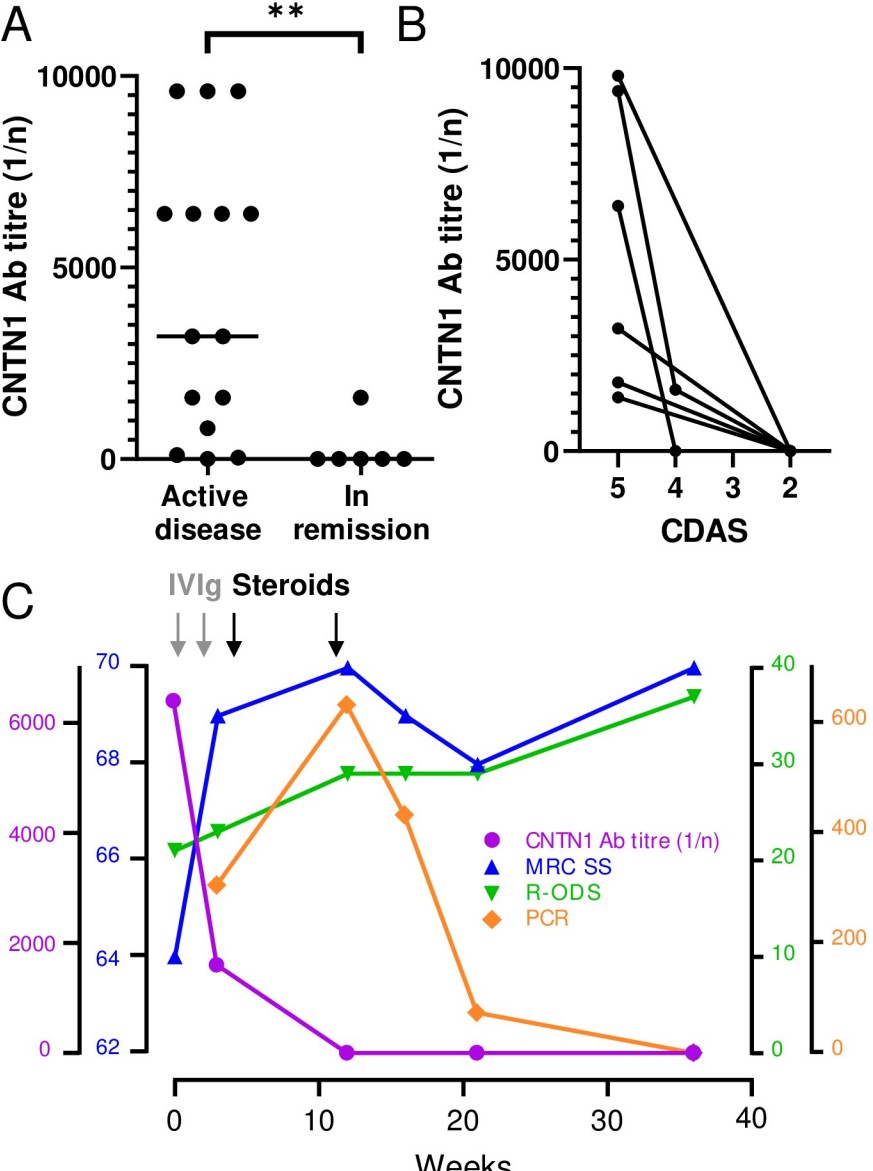

**Fig 3. CNTN1 antibody titres correlate with disease activity. A)** CNTN1 antibody titres were significantly higher in active disease (on or pre-treatment) (n = 15), compared to remission off treatment) (n = 6, p = 0.001, Mann-Whitney). **B)** In all 6 patients for whom serial samples were available, CNTN1 antibody titres fell in parallel with a reduction in neuropathy disease activity (measured using the CIDP Disease Activity Scale–CDAS) and became undetectable in 5 patients who achieved treatment-free remission. **C)** In one patient, where this level of detail was available, serial assessments showed that improvements in strength (Medical Research Council sum score, MRC-SS), functional abilities (inflammatory neuropathy Rasch-built Overall Disability Scale, iR-ODS) and normalisation of renal function (Protein Creatinine Ratio, PCR) correlated with falling CNTN1 antibody titres. Grey arrows indicate IVIg administration (2g/kg over 5 days), and black arrows pulsed dexamethasone (40mg daily for 4 days, repeated after 4 weeks).

basement membrane of an affected patient with MGN [30]. This suggests that, similar to what is seen in anti-PLA$_2$R MGN, circulating CNTN1 antibodies bind directly to the antigen *in situ*, which is then shed and accumulates in the glomerular space [31].

Emerging data also reveals the presence of CNTN1 protein in serum, and show levels reflect antibody status and neurological disease activity [18]. In contrast to PLA$_2$R, which is

undetectable in the serum of either controls or anti-PLA$_2$R MGN patients [7], we detected CNTN1 protein in the serum of healthy controls, as well as seronegative CIDP and other forms of nephrotic syndrome, but levels were significantly lower in CNTN1 antibody-positive patients (Fig 2). The detection of CNTN1 protein within the PEG-precipitated fraction of patient serum sampled during a period of high anti-CNTN1 antibody titre suggests that CNTN1 may exist as circulating immune complexes within the blood in CNTN1 antibody-positive patients. The lack of mesangial and/or capillary immune complexes in patient biopsies may argue against these being the cause of nephrotic syndrome [32], though it remains possible that smaller circulating immune complexes can cause a subepithelial pattern of deposition [33]. Our findings encourage a fresh look at the potential for deposition of circulating immune complexes in human renal disease [34].

Recent studies provide insight into the pathogenic mechanisms of CNTN1 antibody-mediated peripheral neuropathy [35, 36]. Why some CNTN1 antibody patients preferentially develop nerve or renal injury is unclear. One or other disease may simply remain sub-clinical. Alternatively, pre-existing kidney injury may facilitate subsequent immune-mediated glomerular pathology, as can be seen in mice, where CNTN1 expression is upregulated within the kidney by glomerular injury [37]. The prior diagnosis of diabetes in over a quarter of patients in our cohort may be relevant in this regard. Further, antibodies against glomerular antigen THSD7A can be associated with an increased risk of cancer-associated MGN [38]. Although CNTN1 has well established roles in association with numerous solid organ cancers [39], malignancy was not detected as a potential source of CNTN1 antigen in the 6 patients in our cohort who were radiologically screened (S1 Table in S1 File). However, it is possible some patients may have subsequently developed cancer following diagnosis, as has been previously described [40].

In summary, the improvement of neuropathy and renal disease following immunotherapy highlights the utility of CNTN1 antibodies in the identification of a clinically reversible disorder.

## Supporting information

**S1 Checklist.**
(DOCX)

**S1 Fig. Flowchart summarising detection of patients with CIDP, nephrotic syndrome and CNTN1 antibodies through prospective nodal/paranodal autoantibody screening.**
(TIFF)

**S2 Fig. Titre and subclass specificity comparison between CBA and ELISA.**
(TIFF)

**S3 Fig. Immunoreactivity of IgG from CNTN1-positive patients on CBA show they are specific for Contactin-1 (CNTN1).**
(TIFF)

**S4 Fig. Axonal pattern of IgG binding from CNTN1-positive patients in myelinated neuronal co-cultures is specific to the CNTN1 antigen.**
(TIFF)

**S5 Fig. Contactin-1 immunoreactivity in kidney immune deposits for patents 1, 2 and 13.**
(TIFF)

**S6 Fig. LC-MS/MS of CNTN1 peptides.**
(TIFF)

**S7 Fig. CNTN1 is expressed in healthy human kidney.**
(TIFF)

**S8 Fig. CNTN1-containing immune complexes are present in serum at disease nadir.**
(TIFF)

**S1 File.**
(DOCX)

**S1 Raw images.**
(PDF)

## Acknowledgments

We thank Prof. Moin Saleem (University of Bristol) for the kind gift of the human podocyte cultures, and Dr Lucy McDermott for the *YWHAZ* PCR primers. We thank Hannah Brooks and acknowledge the Oxford Brain Bank, supported by the Medical Research Council (MRC), Brains for Dementia Research (BDR) (Alzheimer Society and Alzheimer Research UK), Autistica UK and the NIHR Oxford Biomedical Research Centre for frozen human brain tissue. We also thank Dr David Maldonado-Perez and the team at the Oxford Centre for Histopathology Research (OCHRe) and Oxford Radcliffe Biobank (ORB) supported by the University of Oxford, the Oxford Biomedical Research Centre (BRC) Digital Pathology theme, Cancer Research UK (CRUK) Oxford Cancer Centre, National Institute for Health Research Thames Valley and South Midlands Clinical Research Network (NIHR CRN) and Research Capability Funding (NIHR RCF) for the frozen human kidney samples. The MRC GN bank was supported by the MRC and Kidney Research UK.

## Author Contributions

**Conceptualization:** Alexander J. Davies, Alan D. Salama, Simon Rinaldi.

**Data curation:** Alexander J. Davies, Simon Rinaldi.

**Formal analysis:** Janev Fehmi, Alexander J. Davies.

**Funding acquisition:** Simon Rinaldi.

**Investigation:** Janev Fehmi, Alexander J. Davies, Sonja Pikkupeura, Ruth Pepper, Filipa Farinha, Anisur Rahman, Diana Canetti, Janet A. Gilbertson, Nigel B. Rendell, Filip Eftimov, Luuk Wieske, Charlotte E. Teunissen, Ian S. D. Roberts, Neil Ashman, Alan D. Salama, Simon Rinaldi.

**Methodology:** Janev Fehmi, Alexander J. Davies, Alan D. Salama, Simon Rinaldi.

**Project administration:** Janev Fehmi, Alan D. Salama, Simon Rinaldi.

**Resources:** Marilina Antonelou, Luis Querol, Emilien Delmont, Andrea Cortese, Diego Franciotta, Staffan Persson, Jonathan Barratt, Aleksandar Radunovic, Thomas Minton, Geraint Fuller, Sinead M. Murphy, Aisling S. Carr, Mary R. Reilly, Ian S. D. Roberts, Neil Ashman, Alan D. Salama, Simon Rinaldi.

**Supervision:** Alan D. Salama, Simon Rinaldi.

**Validation:** Janev Fehmi, Alexander J. Davies, Simon Rinaldi.

**Visualization:** Janev Fehmi, Alexander J. Davies, Alan D. Salama, Simon Rinaldi.

**Writing – original draft:** Janev Fehmi, Alexander J. Davies, Alan D. Salama, Simon Rinaldi.

**Writing – review & editing:** Janev Fehmi, Alexander J. Davies, Marilina Antonelou, Stephen Keddie, Sonja Pikkupeura, Luis Querol, Emilien Delmont, Andrea Cortese, Diego Franciotta, Staffan Persson, Jonathan Barratt, Ruth Pepper, Filipa Farinha, Anisur Rahman, Diana Canetti, Janet A. Gilbertson, Nigel B. Rendell, Aleksandar Radunovic, Thomas Minton, Geraint Fuller, Sinead M. Murphy, Aisling S. Carr, Mary R. Reilly, Filip Eftimov, Luuk Wieske, Charlotte E. Teunissen, Ian S. D. Roberts, Neil Ashman, Alan D. Salama, Simon Rinaldi.

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
