## [Decision Letter · Decision Letter 0]

16 Jan 2023

Contactin-1 links autoimmune neuropathy and membranous glomerulonephritis

PONE-D-22-31361

Dear Dr. fehmi,

We’re pleased to inform you that your manuscript has been judged scientifically suitable for publication and will be formally accepted for publication once it meets all outstanding technical requirements.

Kind regards,

Zhanjun Jia

Academic Editor

PLOS ONE

1. Please provide additional details regarding participant consent. In the ethics statement in the Methods and online submission information, please ensure that you have specified (1) whether consent was informed and (2) what type you obtained (for instance, written or verbal, and if verbal, how it was documented and witnessed). If your study included minors, state whether you obtained consent from parents or guardians. If the need for consent was waived by the ethics committee, please include this information.

"I have read the journal's policy and the authors of this manuscript have the following competing interests: 

SR runs a not-for-profit diagnostic testing service for nodal/paranodal antibodies. 

All other authors have declared that no competing interests exist."

Please respond by return email with your amended Competing Interests Statement and we will change the online submission form on your behalf.

Additional Editor Comments:

This is a well performed research work and the MS is well prepared.

Reviewers' comments:

Reviewer's Responses to Questions

**Comments to the Author**

1. Is the manuscript technically sound, and do the data support the conclusions?

Reviewer #1: Yes

2. Has the statistical analysis been performed appropriately and rigorously? 

Reviewer #1: Yes

3. Have the authors made all data underlying the findings in their manuscript fully available?

Reviewer #1: Yes

4. Is the manuscript presented in an intelligible fashion and written in standard English?

Reviewer #1: Yes

5. Review Comments to the Author

Reviewer #1: This is a very interesting finding, and all the data are solid for the conclusion. Those studies indicate that Contactin-1 (CNTN1) in peripheral nerves and kidney glomeruli may act as diagnostic markers for autoimmune neuropathy and membranous glomerulonephritis. This manuscript provides new insights into clinical studies. One minor revision suggestion is to enlarge the label in Figure S3 as they are unclear.

6. PLOS authors have the option to publish the peer review history of their article (what does this mean?). If published, this will include your full peer review and any attached files.

Reviewer #1: No

---

## [Editor Report · Acceptance letter]

27 Feb 2023

PONE-D-22-31361 

Contactin-1 links autoimmune neuropathy and membranous glomerulonephritis 

Dear Dr. fehmi:

I'm pleased to inform you that your manuscript has been deemed suitable for publication in PLOS ONE. Congratulations! Your manuscript is now with our production department. 

Kind regards, 

on behalf of

Dr. Zhanjun Jia 

Academic Editor

PLOS ONE